# Ribosomal RNA Transcription Regulation in Breast Cancer

**DOI:** 10.3390/genes12040502

**Published:** 2021-03-29

**Authors:** Cecelia M. Harold, Amber F. Buhagiar, Yan Cheng, Susan J. Baserga

**Affiliations:** 1Department of Genetics, Yale University, New Haven, CT 06510, USA; cecelia.harold@yale.edu (C.M.H.); yan.cheng@yale.edu (Y.C.); 2Department of Molecular Biophysics and Biochemistry, Yale University, New Haven, CT 06510, USA; amber.buhagiar@yale.edu; 3Department of Therapeutic Radiology, Yale University, New Haven, CT 06510, USA

**Keywords:** ribosome biogenesis, nucleolus, RNA Polymerase I, rRNA synthesis, breast cancer

## Abstract

Ribosome biogenesis is a complex process that is responsible for the formation of ribosomes and ultimately global protein synthesis. The first step in this process is the synthesis of the ribosomal RNA in the nucleolus, transcribed by RNA Polymerase I. Historically, abnormal nucleolar structure is indicative of poor cancer prognoses. In recent years, it has been shown that ribosome biogenesis, and rDNA transcription in particular, is dysregulated in cancer cells. Coupled with advancements in screening technology that allowed for the discovery of novel drugs targeting RNA Polymerase I, this transcriptional machinery is an increasingly viable target for cancer therapies. In this review, we discuss ribosome biogenesis in breast cancer and the different cellular pathways involved. Moreover, we discuss current therapeutics that have been found to affect rDNA transcription and more novel drugs that target rDNA transcription machinery as a promising avenue for breast cancer treatment.

## 1. Introduction

Breast cancer (BC) has become a global health burden with the second highest incidence in combined sexes—it is the most frequent malignancy in women. Based on the statistical report from the International Agency for Research on Cancer, BC affected around 2.1 million individuals and 630,000 females died of BC in 2018 [1]. Younger women diagnosed with BC usually have the highest local recurrence rates [2]. Experts believe the global incidence and mortality rates will keep increasing in the next few years, marking a need for continued research on BC and treatment options [2].

The management of BC depends on stages and molecular features. With current therapeutic options, 70–80% of patients with early-stage BC are considered curable. In contrast, while advanced BC is treatable, there is a critical need for new therapies for metastatic disease [3]. According to the presence/absence of estrogen or progesterone receptors (ER, PR), and amplification of human epidermal growth factor receptor 2 (HER-2, encoded by *ERBB2*), BC is categorized into three main subtypes: ER+PR+/HER2− (70%), HER+ (15–20%), and triple-negative (15%, TNBC) [4]. Among the three subtypes, triple-negative BC is the deadliest, having the lowest 5-year survival rate for stage I (85%), and the median overall survival for metastatic triple-negative BC is 1 year, which is just one-fifth of the survival for the other two subtypes of metastatic BC [4,5]. The subtype of BC has a great impact on deciding treatment strategy. Endocrine therapy and targeted therapy, sometimes combined with chemotherapy, are typically applied in systemic treatment for ER+PR+/HER2− and HER+ BC. Patients with TNBC usually receive chemotherapy alone [4]. The intertumor and intratumor heterogeneity of BC also lead to chemo-resistance, which is one of the main obstacles in BC management [6]. Currently, there is still an urgent need for new therapeutic strategies in treating BC, especially for TNBC and metastatic BC [7].

Ribosome biogenesis (RiBi) has become an attractive druggable pathway in recent years. Pianese et al. first observed that cancer cells had larger nucleoli than normal cells in 1896 [8], which put forward the connection between malignancy and nucleoli, the main cellular location for ribosomal RNA (rRNA) synthesis. Though enlarged nucleoli are not usually associated with BC cells [9], as shown in Figure 1, the morphometric alteration of nucleoli is still an important prognostic parameter for invasive BC to assess tumor aggressiveness [10,11]. Prominent nucleoli are usually associated with poor prognosis in BC [11]. Nucleolar alteration can be an indicator of hyperactive RiBi, which is needed for unchecked cell proliferation in cancer [12]. Recently, there has been growing evidence suggesting the roles of stimulated RiBi in tumorigenesis [9,13,14]. For example, upregulated RiBi attenuates the expression and activity of p53, a well-known tumor suppressor, which can lead to neoplastic transformation [9]. Meanwhile, oncogenes and the loss of tumor suppressor genes can stimulate ribosome production [12,15]. Specifically, in BC, enhanced tumor aggressivity is associated with increased pre-rRNA synthesis [16]. Thus, inhibiting RiBi has become a promising strategy in BC treatment, and rRNA transcription, the initial step of RiBi has become the main focus. However, there is no study directly comparing the ribosome biogenesis rate in different subtypes of BC, which leaves a gap in this field.

In this review, we will first summarize the process of ribosome biogenesis and introduce the nucleolar adaptation that occurs in breast cancer. We will then focus on RNA Polymerase I (RNAPI) transcription, and further discuss the different signal transduction pathways that modulate RNAPI activity. Finally, we discuss both the new and old drugs targeting rRNA synthesis that have potential application in BC treatment.

## 2. Ribosome Biogenesis

### 2.1. Ribosome Biogenesis

The nucleolus is a highly dynamic, membraneless organelle that is critically important for driving ribosome biosynthesis. Nucleoli form around expressed rDNA gene clusters intrinsic to the short arms of human acrocentric chromosomes 13, 14, 15, 21, and 22. These sites are termed nucleolar organizer regions (NORs) [17,18]. After formation, the nucleolus is structurally organized into three concentric sub-compartments that facilitate different steps of RiBi: the fibrillar center (FC), the dense fibrillar component (DFC), and the granular component (GC) [19]. RiBi initiates at the interface of the interior FC and DFC with the transcription of the rDNA genes by RNAPI to produce the 47S pre-ribosomal (pre-rRNA) transcript (Figure 2). The 47S pre-rRNA transcript is further processed and chemically modified in the surrounding DFC to give rise to the mature 18S, 5.8S, and 28S rRNAs. A fourth rRNA species, the 5S rRNA, is transcribed from chromosome 1 by RNA polymerase III (RNAPIII) in the nucleoplasm. In addition to ~200 *trans*-acting factors including small nucleolar RNAs (snoRNAs), assembly factor proteins and ribosomal proteins transcribed by RNA polymerase II (RNAPII) help modulate ribosomal processing and assembly throughout this process, and coalesce with the mature rRNAs in the peripheral GC to form the small pre-40S (18S rRNA) and large pre-60S (5.8S rRNA and 28S rRNA, and the 5S rRNA) subunits. After exiting the nucleus, the final maturation steps take place in the cytoplasm and the 40S and 60S subunits join to perform protein translation.

The number of ribosomes in a cell dictates translational capacity, and protein synthesis underpins a number of fundamental cellular processes including cell growth and proliferation. Since ribosome production depends on the fidelity of rRNA synthesis, ensuring faithful rDNA transcription is key for maintaining cellular homeostasis. As such, transcription of the rDNA by RNAPI represents approximately 60% of nuclear transcription and the genes at rDNA loci are organized in tandem repeat units to satisfy the high demand for rRNA generation [20]. There are approximately 300 rDNA repeats in a human diploid cell, and only a subset of these gene units are active for transcription at any given time [21,22]. Interestingly, rDNA copy number is variable in metastatic breast cancer cells where both loss/gain events have been observed [23]. rDNA transcription requires the assembly of critical factors along the promoter start site called the pre-initiation complex (PIC). First, upstream binding factor (UBF) binds to the upstream control element (UCE) [24]. From there, UBF mediates the recruitment of the SL1 complex to the core promoter [25]. Finally, SL1 recruits the RNAPI holoenzyme to the promoter via interaction with auxiliary factor TIF-1A (also known as RRN3) initiating transcription of the rDNA genes [26]. Dysregulation of this process is implicated in a wide range of human diseases and disorders including Treacher-Collins syndrome, a more well-characterized ribosomopathy, and cancer [27].

### 2.2. Nucleolar Remodeling in Breast Cancer

The nucleolar ultrastructure is a barometer for the overall health of the cell, and is often altered in response to genetic and environmental changes. For example, our laboratory demonstrated that RNAi-mediated knockdown of protein regulators, including known RiBi factors, distorts nucleolar number in MCF10A breast epithelial cells [28,29]. Notably, aberrant nucleolar morphology has been long associated with cancer cells where nucleoli are enlarged and irregularly shaped in malignancy [8,30]. Advanced cytochemical procedures have enabled visualization of nucleolar organizer regions in tumor tissues by silver staining associated argyrophilic proteins (AgNOR) [31,32]. Studies measuring AgNOR have shown that increased nucleolar size, or nucleolar hypertrophy, mirrors increased RNAPI transcription and thus higher ribosome production to meet the metabolic requirements of proliferating cancer cells [33]. However, in malignant cancerous tissues where cells are slow dividing, the nucleolus structure often remains unaffected reducing the diagnostic utility based on nucleolar hypertrophy alone [34]. Nonetheless, the link between cancer and nucleolar adaptation cannot be ignored and nucleolar morphometry is more often employed by pathologists as a prognostic parameter of breast cancer.

Recently, Elsharawy and colleagues assessed nucleolar size alterations in hematoxylin and eosin-stained tumor cells extracted from 1600 patients with invasive breast cancer by assigning higher nucleolar scores to tumor cells displaying prominent nucleoli. It was found that higher nucleolar scores were associated with poorer prognosis and other clincopathological parameters including younger patient age and stage 3 [11]. Similar observations have been observed in a previous AgNOR-based study in which breast cancer tissues’ mean nucleolar area was evaluated and smaller nucleolar areas were shown to correlate with stronger survival rates in patients [35]. Since rDNA transcription drives nucleolar formation and is the rate limiting step for cellular proliferation, advances in our understanding of molecules and pathways involved in controlling RNAPI activity will lead to new insights into the pathogenetic mechanisms underlying breast cancer tumorigenesis.

## 3. Signal Transduction Pathways Modulate RNAPI Activity

### 3.1. MYC

The transcription factor c-MYC is a downstream target of many signaling pathways and plays a critical role in the RNAPII-driven transcription of genes involved in cell cycle progression, ribosome biogenesis, and cell adhesion [36,37]. The activity of the MYC transcriptional interactome is frequently deregulated in breast cancer and contributes to mammary tumor development and progression. For example, in several transgenic mice studies, c-Myc overexpression in the mammary gland induced tumor formation [38,39]. A recent meta-analysis investigating the relationship between c-Myc expression and prognostic outcome in breast cancer found that high c-Myc expression was associated with increased tumor size and poor relapse free survival [40]. Among target genes, MYC can influence the expression of specific RNAPI transcription components including TIF-1A (RRN3), UBF, and RNAPI subunits POLR1B and POLR1E thereby promoting rDNA transcription (Figure 2) [41,42]. In addition, c-Myc can also enhance ribosome biogenesis by regulating RNAPIII transcription activity and the expression of ribosomal proteins [36]. Moreover, c-Myc facilitates the formation of the pre-initiation complex by binding to the rDNA promoter and mediating the recruitment of the SL1 complex to directly initiate RNAPI activation and thus transcription of the rRNA genes [43,44]. c-Myc has also been linked to maintenance of the nucleolar structure and its over-expression results in a significant enlargement of nucleoli [45,46].

### 3.2. PI3K/AKT/mTOR

The PI3K/AKT pathway controls a broad spectrum of cellular activities including the regulation of cell growth, metabolism, and autophagy. Hyperactivation of this pathway is implicated in the development of breast cancer and frequently results in the upregulation of its main downstream effector, the mechanistic target of rapamycin (mTOR) [47,48]. mTOR can mediate cellular proliferation by potentiating rRNA synthesis and responds to different environmental signals, including growth factors and nutrient availability, to regulate rDNA transcription through several mechanisms. mTOR can aid PIC assembly via its downstream target ribosomal protein S6 kinase 1 (S6K1) which phosphorylates the C-terminal activation domain of UBF. This interaction in turn promotes rDNA promoter occupancy of SL1 [49]. Similarly, mTOR signaling can modulate the activity and localization of basal component TIF-1A by catalyzing the phosphorylation of residues S44 and S199 to activate or inactivate TIF-1A, respectively [50]. Moreover, mTOR can increase rRNA synthesis by directly binding to the rDNA promoter and indirectly by inducing MYC translation [51,52]. Independently of mTOR, AKT can also indirectly regulate TIF-1A via CK2 which phosphorylates residues S170 and S172, augmenting TIF-1A binding and RNAPI activity at the rDNA locus [53].

TIF-IA (also known as RRN3) is an important rate-limiting factor of rDNA transcription initiation and is a target shared by the c-Myc and mTOR oncogenic pathways. Not surprisingly, after screening a series of breast cancer cell lines, Rosetti and colleagues detected significant TIF-IA upregulation with a concomitant elevation in levels of pre-rRNA. The authors also showed that overexpressing TIF-IA in human mammary epithelial cells (HME-1) was enough to increase rRNA synthesis and disrupt 3D epithelial morphogenesis. The same study also analyzed the presence of genomic alterations in factors associated with the RNAPI transcription machinery in early breast cancer lesions and invasive breast carcinomas. Remarkably, their analysis found frequent upregulation in one or more of these genes in both early and advanced stages of breast cancer including frequently TIF-IA [54]. This suggests that the induction of rRNA synthesis in breast cancer cells is driven by the increased expression of associated RNAPI transcription factors, and may be a key contributor in the initiation of mammary oncogenesis.

### 3.3. Wnt

Wnt signaling encompasses a family of important pathways that regulate cell division and differentiation in tissue morphogenesis. The canonical Wnt/β-catenin signaling pathway is important for mammary gland development and the aberrant regulation of this pathway is implicated in mammary oncogenesis [55]. Wnt/β-catenin signaling indirectly affects RNAPI activity by controlling the expression of rDNA transcription associated factors including c-Myc and putative regulator PPAN (Figure 2) [56]. In non-basal like breast cancer cells, β-catenin was shown to upregulate c-Myc expression [57]. Recently, Weeks et al. reported that TNBC cell lines with upregulated Wnt/β–catenin signaling feature an increased number of nucleoli per cell nucleus. Further, treating cells with an inhibitor of beta-catenin driven transcription led to a significant reduction in nucleolar number suggesting decreased rDNA transcription and thus reduced ribosome biogenesis [58]. Interestingly, PPAN was recently identified as a novel prognostic marker of TNBC where high expression of PPAN was associated with poor overall survival [59]. PPAN was previously shown to localize to the nucleolus and stabilizes UBF, aiding in 47S rRNA maturation [60]. Additionally, it was recently demonstrated that over-activation of the canonical Wnt pathway in response to nucleolar stress in cancer cells stimulates PPAN expression to maintain rRNA synthesis [61].

Conversely, non-canonical, β-catenin independent Wnt signaling negatively regulates rDNA gene transcription in breast cancer. Reduced protein levels of the non-canonical Wnt ligand, Wnt5a, is observed in 45–75% of breast cancer patients and is associated with poor prognosis including early relapse and reduced disease free-survival. Wnt5a is thought to antagonize tumor growth by impairing cell migration affecting proliferative capacity [62]. Formerly, Dass et al. showed that transgenic mice expressing Wnt5a null mammary tumors expressed higher levels of the proliferation marker Ki-67 and increased AgNORs compared to wild-type tumors [63]. In addition, treating human MCF7 breast cancer cells with recombinant Wnt5a led to a significant reduction in rRNA synthesis as measured by 47S pre-rRNA levels, cellular proliferation, and decreased nucleolar area. The authors demonstrated that these results were affected by Dishevelled1 (DVL1), a downstream target of Wnt5a, which binds to the rDNA gene promoter and induces the dissociation of the deacetylase SIRT7 from the RNAPI machinery [63]. PAF53, a subunit of RNAPI, is deacetylated by SIRT7 which enhances the associative capability of RNAPI with the rRNA genes [64]. Therefore, the crosstalk between both canonical and non-canonical Wnt signaling and rRNA transcription can influence breast tumorigenesis.

## 4. Tumor Suppressor Proteins Inhibit rDNA Transcription

### 4.1. p53 & pRb

Two tumor suppressor proteins, p53 (Tumor protein p53) and pRb (retinoblastoma protein) exhibit similar functional roles including cell cycle regulation by inducing arrest at the G1/S border in response to DNA damage and cellular stress [65,66]. These factors are frequently mutated in breast cancers, notably basal-like breast cancer, where p53 and pRb loss of function is linked to increased proliferation and more aggressive breast carcinomas [67]. In addition to regulating cell cycle progression, p53 and pRb can restrict cell growth and proliferation by repressing rRNA synthesis. This repression is achieved by interfering with PIC formation. p53 and pRb have been demonstrated in vivo and in vitro to inhibit RNAPI transcription by associating with the TBP and TAF(1)48 subunits of the SL1 complex, respectively, which prevents the interaction between UBF and SL1 [68,69,70,71,72]. Additionally, in two AgNOR-based studies evaluating p53 and pRb expression in breast carcinomas, p53 and pRb loss was associated with increased AgNOR mean area and tumors with high AgNOR values were characterized by worse prognosis [73,74].

In addition to being the main site of RiBi, the nucleolus is also sensitive to cellular stressors, such as heat shock or hypoxia. In reaction to stressors, RiBi is perturbed and initiation of the nucleolar stress response (NSR) occurs, which can cause cell cycle arrest and apoptosis. The NSR causes cell cycle arrest through the release of ribosomal proteins and the 5S ribonucleoprotein (5S-RNP) that bind to and inhibit the E3-ubiquitin ligase MDM2, which normally suppresses p53 levels, and thus allows for an accumulation of p53 [75]. Pestov et al. showed that p53-dependent cell cycle arrest and inhibition of RiBi occurred through expression of a dominant negative form of Bop1, part of the PeBoW complex responsible for proper rRNA cleavage, clearly linking p53, RiBi, and the cell cycle [76]. Rubbi and Milner established that nucleolar stress causes p53 stabilization [77]. In fact, following irradiation, DNA damage alone was not sufficient to stabilize p53—nucleolar disruption is necessary [77]. Indeed, in cases of ribosomopathies, there is also evidence of p53 mediating clinical manifestations and that depletion of p53 can rescue such effects [78,79]. Nucleolar stress can also be p53-independent, specifically in yeast which lacks MDM2 and p53, reviewed previously [80]. Thus, in cancers where p53 is mutated or absent, targeting nucleolar activity is still a viable option.

### 4.2. PTEN

The *PTEN* gene (phosphatase and tensin homolog) encodes a tumor suppressor protein that functions as a lipid phosphatase. *PTEN* directly and indirectly affects cell survival, proliferation, and apoptosis and is frequently deleted in breast cancer [81]. *PTEN* can directly mediate rDNA transcription repression by interacting with the RNAPI machinery and directing the dissociation of the SL1 complex [82]. Additionally, *PTEN* negatively regulates the PI3K/AKT/mTOR pathway, thereby indirectly reducing rRNA synthesis and proliferation [83]. To more fully define the prognostic significance of *PTEN* expression in breast tumorigenesis, Li et al. conducted a meta-analysis of 27 publications encompassing 10,231 patients. Their findings showed significant associations between *PTEN* loss and clinicopathological features including larger tumor size and lymph node metastasis. Their results also indicate that *PTEN* loss predicted worse prognosis and was frequently associated with the TNBC [84]. Therefore, sufficient *PTEN* expression is indispensable for maintaining normal cell function and preventing uncontrolled proliferation in part by decreasing transcription of the rRNA genes.

### 4.3. ARF

The p14 alternative reading frame (p14Arf) tumor suppressor protein is an upstream regulator of p53 and has been linked to the regulation of cell-cycle during G2 phase [85]. p14Arf also interacts directly with the rDNA promoter and, upon overexpression, can suppress rDNA transcription by causing dephosphorylation of UBF serine residues S388 and S484 [86,87]. p14Arf can further inhibit rRNA synthesis by restricting the nucleolar localization of the TTF-I termination factor thereby blocking its interaction with the rDNA promoter region [88]. TTF-I facilitates transcription termination and re-initiation by RNAPI by promoting the release of nascent rRNA [89]. Although the role of p14Arf in breast cancer is not well-defined, its expression has been previously reported to be altered in mammary tumors and a recent study analyzing invasive ductal carcinomas from 1080 patients found that overexpression of p14ARF, as opposed to loss, was associated with poor prognosis [90,91]. Interestingly, the authors report that p14Arf expression was primarily observed in the cytoplasm, although the protein accumulates in the nucleolus, suggesting functional inactivation [90]. Nonetheless, this delocalization event would remove an additional block on rDNA transcription imposed by p14Arf, although this has yet to be elucidated experimentally.

## 5. Regulation of rRNA Synthesis by Noncoding RNAs

An added layer of control of rDNA transcription is exerted by non-coding RNA species—specifically microRNAs (miRNAs) and long non-coding RNAs (lncRNAs), reviewed previously [92]. A number of these miRNAs control rRNA synthesis through interactions with MYC and p53, further linking oncogenesis and RiBi. For example, repression of ribosomal protein RPL22 results in the upregulation of LIN28B, an RNA-binding protein that regulates *let-7* miRNA maturation, and subsequent downregulation of the *let-7* family of miRNAs and ultimate activation of MYC, which increases rRNA synthesis [93]. miR-24, miR-130a, and miR-145 repress MYC through interactions with ribosomal proteins RPL5, RPL11, and RPS14, respectively [94,95,96,97]. miR-424-5p was found to target elements of the transcriptional machinery, POLR1A and UBTF, leading to a depletion of rRNA levels and protein synthesis in murine muscle cells (Figure 2) [98]. Finally, miR-504 upregulates the nucleolar protein FGF13 and represses p53, and an increase in this miRNA disrupts rRNA synthesis [99].

lncRNAs are also responsible for the control of rDNA transcription, and can interact directly with RNAPI activity. An increase in levels of the lncRNA, long nucleolar RNA (LoNA), causes a consequent reduction in pre-rRNA and mature rRNA levels [100]. LoNA has two 5’ nucleolin binding motifs that allow for it to bind to and sequester the nucleolar protein nucleolin, prohibiting rDNA euchromatin modifications. Promoter associated RNA (pRNA), transcribed by RNAPI, recruits the nucleolar remodeling complex (NoRC) to silence rDNA transcription [101]. Promoter and pre-rRNA antisense transcripts (PAPAS) respond to stress conditions, regulating chromatin modifications that signal to turn rDNA transcription on and off [102,103,104]. An additional lncRNA, snoRNA-ended lncRNA that enhances pre-rRNA transcription (SLERT), directly interacts with the RNA helicase DDX21 to relieve its inhibition of rDNA transcription [105]. A number of ncRNAs have regulatory implications in RiBi, with discoveries of more novel regulators still being conducted.

## 6. Therapeutic Targeting of rRNA Synthesis to Treat Breast Cancer

The nucleolus is an increasingly viable target for drug intervention in cancer therapeutics. Increased ribosome biogenesis is driven in part by overactivation of known oncogenes such as MYC and the stabilization of the tumor suppressor p53 [106,107]. rRNA synthesis is commonly dysregulated in cancer, solidifying this step as a potential therapeutic target [13,108]. Furthermore, RNAPI is also an emerging target for new antineoplastic drugs [109,110,111,112]. There have been several drugs recently that have shown inhibitory effects on RNAPI activity and thus, RiBi (Figure 3). These drugs are of particular interest in their use as anti-cancer agents, including breast cancer specifically.

### 6.1. Current Anti-Cancer Drugs with Effects on RNAPI Activity

Drugs already in use for cancer treatment have well-documented roles in inhibiting RiBi and nucleolar function. For example, actinomycin D preferentially intercalates with GC-rich DNA, which stalls RNA polymerases at replication forks and halts elongation [113]. Due to rDNA having a higher percentage of GC content, and being highly prevalent in the cell, actinomycin D intercalates with the rDNA, stalling RNAPI transcription. Further, this intercalation also forms stable Topoisomerase I-DNA complexes that block transcription (Topoisomerase I and II are denoted as Top1/2 in Figure 3) [114]. A host of current chemotherapeutics were screened for effects on RiBi in human fibrocarcinoma cells (2fTGH) [115]. These drugs were found to have specific effects on rDNA transcription, early rRNA processing, or late rRNA processing, and a total of 36 drugs were found to have this activity [115]. Three drugs—doxorubicin, methotrexate, and mitoxantrone—had effects specifically on rDNA transcription. All three drug treatments showed a marked decrease in labelled 47S pre-rRNA transcript levels. Doxorubicin, an anthracycline antibiotic used to treat early and metastatic breast cancer, was shown in *S. cerevisiae* to repress 34 genes related to ribosomal small subunit (SSU) biogenesis, assembly, and processing [116]. In HeLa cells, doxorubicin and other similar drugs cause further ribosome maturation and export deficiencies in addition to the transcriptional effects [117]. Methotrexate and mitoxantrone are also used for breast cancer therapy. Methotrexate treatment inhibits folate synthesis and ultimately nucleotide biosynthesis [118].

Another class of drugs that have both anti-cancer uses and effects on RNAPI activity are platinum-containing drugs. This includes cisplatin, which is commonly used to treat breast cancer, and oxaliplatin, which is more typically used for gastrointestinal cancers [119]. Cisplatin treatment results in the formation of cisplatin-DNA adducts that are detected by DNA repair machinery [120]. These DNA adducts attract UBF, an essential transcription factor for rRNA synthesis [121]. The loss of UBF blocks rRNA synthesis and ultimately causes p53-independent apoptosis [121]. Oxaliplatin was recently found to not induce DNA damage, like cisplatin, but does cause RiBi defects, as measured by depletion of the pre-rRNA and an increase in RPL11 expression [119]. A number of current therapies have effects on RNAPI transcription, further supporting the idea of targeting this process.

### 6.2. CX-3543

In addition to well-established cancer drugs, recently discovered drugs have been found based on their effects on rDNA transcription and nucleolar stress. One such drug is CX-3543 (Quarfloxin). An important aspect of proper rDNA transcription is the spacing of RNAPI on the rDNA. This is in part mediated by G quadruplex-rich regions of the rDNA, and their interactions with nucleolar proteins, such as nucleolin. CX-3543 targets RNAPI activity by disrupting the interaction between nucleolin and the G quadruplexes on the rDNA, causing its dissociation from the rDNA [111]. This disruption directly inhibits RNAPI transcription, with a subsequent accumulation of nucleolin in the nucleoplasm, and induces apoptosis. Further, it was shown to be p53-independent, and specific to the interaction between nucleolin and the rDNA, and no other protein-rDNA interactions. Finally, CX-3543 inhibited cell growth over a variety of cancer cell lines [111]. In colon cancer cells and tissues, CX-3543 treatment reduces the expression of MYC and inhibits cell growth [122]. CX-3543 successfully passed Phase I clinical trials and showed good efficacy and moved into Phase II for neuroendocrine carcinomas but was ultimately abandoned due to bioavailability issues (ClinicalTrials.gov (accessed on 10 March 2021) ID: NCT00485966) [123,124].

### 6.3. CX-5461

CX-5461 was originally discovered in a screen for molecules specifically inhibiting RNAPI transcription [109,110]. It was shown to effectively decrease rRNA synthesis with, at the time, no apparent accompanying DNA damage [109]. CX-5461 was found to target SL1 and its interaction with RNAPI, thereby decreasing rRNA synthesis by 40–60% [110]. In murine models of lymphoma and leukemia, treatment with CX-5461 elicited a p53-dependent apoptotic response—specifically, through the nucleolar surveillance pathway (NSP) [125]. Interestingly, p53 is not necessarily required for CX-5461 activity. In lymphoblastic leukemia cells, CX-5461 treatment activates the noncanonical ATM/ATR pathway, inducing apoptosis in a p53-indepent manner [126,127,128]. A recent study has shown a divergent explanation for its mechanism of action [129]. Here, it was found that the mechanism of cell death is through Topoisomerase II poisoning [129]. Mars et al. also found that CX-5461 blocks the release of RNAPI-RRN3 and blocks transcription initiation [130]. Perhaps CX-5461 has multiple mechanisms of action that could depend on cell context—an interesting avenue to explore.

CX-5461 recently passed phase I clinical trial and a 1/2 dose-escalation trial for hematological cancers [131,132]. CX-5461 and CX-3543 were also later shown to induce DNA damage and rely on the BRCA1/2-mediated homologous repair. Thus, these drugs have the potential to treat breast cancers deficient of damage repair pathway members such as BRCA1/2 [133]. Excitingly, this hypothesis was put directly to use. The clinical trial was expanded to include other malignancies, including breast cancers with abnormal BRCA1/2 (ClinicalTrials.gov (accessed on 10 March 2021) ID: NCT02719977). The data collection for this expansion was estimated to be completed by December 31, 2020, with results hopefully forthcoming. This marks an exciting potential next step in the treatment of breast cancer by inhibiting RNAPI transcription.

### 6.4. BMH-21 and Other BMH Molecules

A subset of BMH drugs was discovered in a screen for p53-activation and antitumor activity [112]. A p53 reporter plasmid transfected in A375 melanoma cells was used for the initial screen for p53 activation. One of the lead compounds, BMH-21, was found to not elicit a DNA damage response, a marked difference from others in the screen and other drugs that affect RiBi. In a follow up to determine the mechanism of action for BMH-21, treatment with the drug caused growth inhibition independent of p53 [134]. Using a fluorescent intercalator displacement (FID) assay, BMH-21 was shown to intercalate with rDNA at GC-rich sites. Using qPCR and metabolic labeling, it was determined that BMH-21 treatment caused a subsequent decrease in rRNA synthesis. Furthermore, BMH-21 treatment caused a specific depletion of the largest subunit of RNAPI, RPA194 (POLR1A), but did not affect other RNAPI complex proteins. BMH-9, -22, and -23 were also found to exert similar control over RPA194 and led to proteosome-dependent degradation of the subunit [134]. There are currently no clinical trials testing the efficacy of any of these BMH molecules; however, the development of specific anti-RNAPI drugs could provide new insights into how inhibiting this essential process can be used to treat various cancers.

### 6.5. Combination Drugs: P1-B1

Perhaps the next step in the development of breast cancer treatments, with both existing and new drugs, is the combination of therapies. Interestingly, the Xu group modified cisplatin with an analog of BMH-21, creating the P1-B1 compound, and studied the molecule’s effect on nucleolar stress in cells as a potential therapeutic [135]. In a variety of cells, including WT and cisplatin-resistant A549 (lung) and MCF7 cells (breast epithelial), P1-B1 caused an increase in cytotoxicity, as compared to cisplatin. The molecule also had an affinity for GC-rich DNA regions. In MCF7 cells, P1-B1 localized to the nucleolus and caused a decrease in 47S levels, confirming that it, like its precursor molecules, has an effect on RNAPI transcription. The structural integrity of the nucleolus is also heavily impacted by treatment of P1-B1, with the most marked distribution changes occurring in the cases of nucleolin and RPA194, suggesting a similar mechanism of action of one of its precursors, BMH-21.

## 7. Closing Remarks/Future Directions

RiBi is an essential process for all cells. Nucleolar morphology has been used as an indicator for prognosis in cancer, but more recently, a dependence of cancer cells on rDNA transcription and subsequent RiBi has become evident [13,108]. Furthermore, advancements in screening technologies have allowed for the discovery of novel mammalian RiBi regulators [28]. Such screens also uncovered that drugs currently in use to treat various cancers, including a number that are used in breast cancer treatments, affect RiBi at various steps in the process [115,117]. Given the reliance of cancer cells on increased protein synthesis and proliferation, and that RNAPI predominantly transcribes one transcript, RNAPI is a good target for new cancer treatments, with potentially fewer effects on healthy cells. Targeting the essential process of making ribosomes in the cell nucleolus, and RNAPI activity more specifically, is an increasingly viable approach to discover new modalities to treat cancer. Additionally, studying the difference of ribosome biogenesis rates in different BC subtypes might provide more insight in treating BC by inhibiting RNAPI transcription.

Overall, old and new drugs target both rRNA synthesis and cancer cell growth. More work is necessary to discover mechanisms of action for chemotherapeutics in use for breast cancer. Further, work in the development of the new drugs, and their implementation in clinical trials for cancer treatment, points to the success of targeting rRNA synthesis as a treatment option. Coupled with current screens of drugs that inhibit RiBi, more therapeutic options have yet to be discovered [136,137]. In the last twenty years, an effort from different laboratories has been underway to discover drugs that target the RNAPI machinery [109,110,111,112]. Two drugs successfully made it to clinical trials to treat different cancers, with one recently passing and expanding into Phase II clinical trials. The latter, CX-5461, in its clinical trial expansion, has been tested for efficacy against breast cancer. The data collection period ended recently, and we eagerly await the results to further determine how targeting RNAPI transcription affects breast cancer in humans. Taken together, the field is moving toward discovering more drugs that target RNAPI transcription, and other steps in RiBi, with the goal of yielding new treatment options for breast cancer.

## Figures and Tables

**Figure 1 genes-12-00502-f001:**
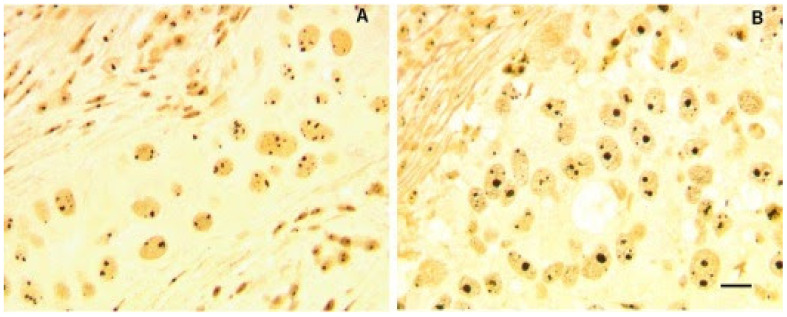
Morphometric alteration of nucleoli in histological sections from two BC samples. Nucleolar proteins are silver stained to visualize nucleoli (black color). Nucleoli in sample (**A**), with wild type p53 and hypo-phosphorylated pRb, showed small size of nucleoli with heterogeneous number per cell. Sample (**B**) was characterized by mutated p53 and deleted RB, having a higher proportion of enlarged nucleoli. Bar = 10 μm. Reproduced with permission from “Ribosome biogenesis and cancer” [9].

**Figure 2 genes-12-00502-f002:**
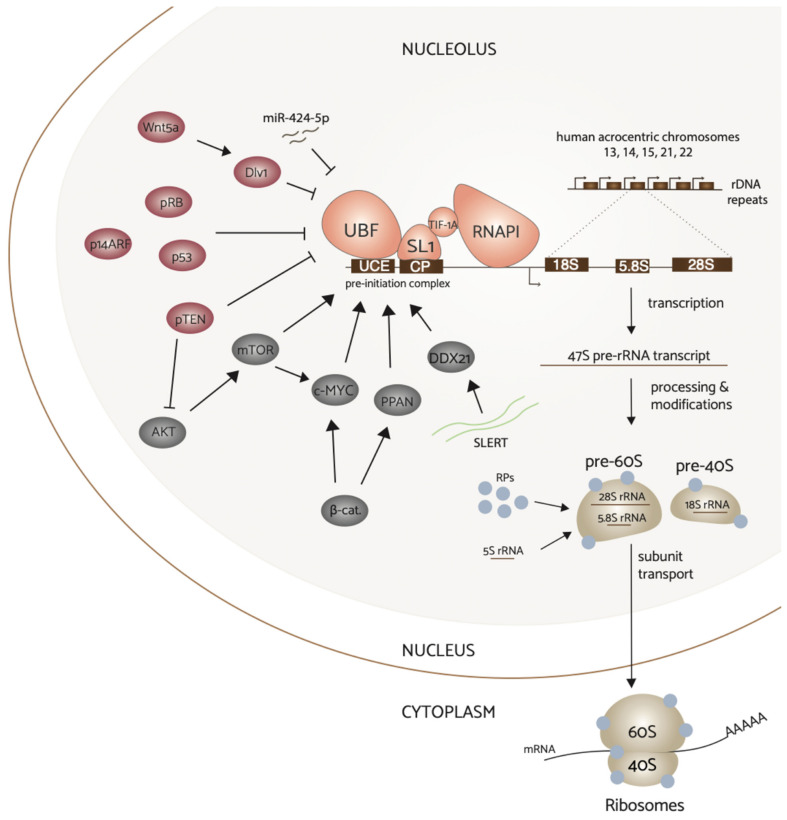
Complex regulation of RNA polymerase I transcription in breast cancer. Hyperactivation of upstream signaling pathways PI3K/AKT/mTOR, Wnt/beta-catenin, and c-Myc converge on and upregulate RNAPI transcription activity. In contrast, non-canonical Wnt signaling via Wnt5a and tumor suppressor proteins p53, pRB, pTEN, and pARF downregulate rDNA transcription. Additionally, ncRNAs miR-424-5p and SLERT can regulate this process. Attenuation of tumor suppressors by inactivation frequently results in elevated rRNA synthesis, and thus increased ribosome production to sustain aberrant proliferation in breast cancer. Ribosome biogenesis (on right) begins with the transcription of the rDNA by RNAPI that yields the 47S pre-rRNA transcript. The transcript is subsequently processed and assembled with ribosomal proteins and the mature 5S rRNA to form the pre-40S and pre-60S subunits. These subunits combine in the cytoplasm to perform protein synthesis.

**Figure 3 genes-12-00502-f003:**
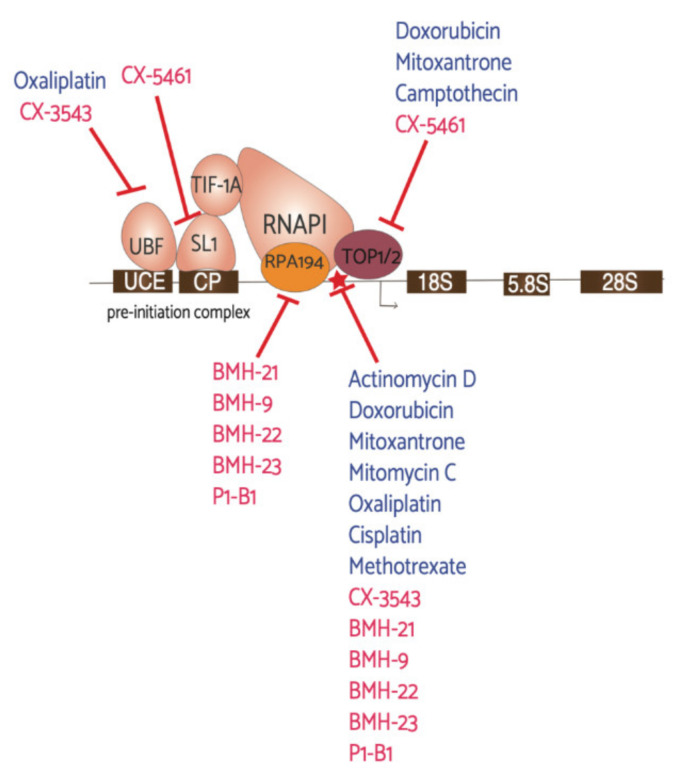
Antineoplastic agents and their RNAPI machinery targets. This schematic represents the first step in ribosome biogenesis, rDNA transcription, and the drugs found to affect this process. Denoted in blue are “older” drugs already in use for cancer treatments and denoted in red are the newly discovered drugs that have been in recent clinical trials, with the exception of the BMH and P1-B1 molecules. The red star highlights direct interaction with the rDNA.

## Data Availability

No new data were created or analyzed in this study. Data sharing is not applicable to this article.

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
