# Peer review of "Ribosomal RNA Transcription Regulation in Breast Cancer"

_genes, 2021, doi:10.3390/genes12040502_

Round 1
Reviewer 1 Report
This is a very nicely written review. My comments:
- Its not clear to me why the authors chose to focus on breast cancer. The cancer angle is pretty general and it could easily be all cancers.
- This is a useful collection of facts, but the authors could do more to highlight exciting new areas or controversies. The review also sounds like everything is well understood.
- On the point of controversies, the authors avoid the topic of phase separation, which is notable in its complete absence. It’s the authors choice whether to include this topic but they should consider mentioning it.
- Nucleolar enlargement in cancers may not derive solely from ribosome biogenesis on overdrive. For example, breast cancers are typically quite aneuploid. Could the authors comment on whether other factors may lead to enlarged nucleoli, e.g. more rDNA chromosomes, proteotoxic stress.
- Figure 1 is copied from Derenzini review paper that is often cited when talking about the nucleolus in cancer. On a closer look it shows that 1- the nuclear size in cancer sample is increased also, 2- there is lots of heterogeneity, 3- there is nothing known about this sample (mutations? what pathways are affected? What is the ploidy?). It is a pathology specimen and nothing is quantified. If the authors could dig up this information it would be really useful for the field, or potentially use a better characterized pathology specimen?
- The Moss lab has a new article on the mechanism of action of CX-5461 that would be good to include in the section on this drug (see NAR Cancer, Mars et al., 2020)
- Please explain what “BMH family” refers to, line 399
- Could the authors please further explain why they think the new Pol1 inhibitors will be a better therapeutic option.
- Some of the sections that are very dense with results from the literature would benefit from a conclusion sentence (e.g. wnt section, PTEN section, ncRNA).
I found a few typos:
- Line 279, delete second “p14Arf” and change interacts to interact
- Line 313, responds should be respond
- The font in figure 3 looks weird, maybe switch to a different font?
Reviewer 2 Report
This is a timely review on ribosomal DNA transcription by RNA Polymerase I and subsequent ribosome biogenesis (RiBi) in human breast cancer cells. The authors do an excellent job in presenting the Pol I transcription machinery and the various signaling mechanisms that regulate rDNA transcription. Most interesting are the updates regarding the various chemical inhibitors for Pol I transcription and their emerging use as chemo-therapeutic agents in cancer treatment.
I have only minor comments and questions:
1) In the second paragraph of the Introduction (lines 33-46), are there any known differences in Pol I transcription rates in ER+PR+/HER-, HER+, or TNBC subtypes?
2) In lines 86 and 88, the authors define RNAPIII and RNAPII respectively, but do not define RNAPI back in line 62.
3) In line 99 of the Introduction, is the number of 300 rDNA repeats referring to the diploid genome, or the haploid genome? And does this number remain constant for breast cancer cells? In other words, do these cancer cells show differences in ploidy and thus (possibly) rDNA copy numbers?
4) Describing cMYC in lines 149-166, besides genes encoding Pol I transcription factors, it might be worth mentioning that other cMYC transcriptional target genes include those encoding downstream ribosome biogenesis factors and the ribosomal proteins themselves. In other words, cMYC is stimulating the entire RiBi system.
5) Line 184: TIF-IA (also known as RRN3)…
6) Line 232: is that supposed to be “Tumor suppressor proteins p53 (p53) and retinoblastoma protein (pRb)…”
7) Line 279: suggesting “p14Arf can also interact directly…”
8) Line 298: perhaps define LIN28B.
9) In Figure 3, does the red star denote interaction with DNA?
10) Lines 373-375: bioavailability of tissues? patients? Did they give up on CX-3543? That’s how I read it. What is NCT00485966?
11) Line 380: “…thereby decreasing rRNA synthesis [106].” Can the authors provide to what extent transcription is blocked by CX-5461?
12) Line 385: p53-independent.
13) Lines 395-396: “…completed by December 31, 2020,…”
14) Perhaps move lines 428-433 into “Closing Remarks”.
15) Future Directions: While much of the review focuses on RNAPI and rDNA transcription, perhaps the authors could propose future efforts that target downstream pre-rRNA processing/ribosome assembly events in cancer cells. Inhibitors that specifically block pre-rRNA processing components (e.g., the U3 snoRNP) could be employed independently or in conjunction with say CX-5461 to block RiBi now at multiple levels. The authors very briefly mention the possibility in line 340.
Reviewer 3 Report
The synthesis of rRNA by RNA polymerase I (Pol I), and maturation of rRNA play a central role in the complex network that controls cell growth and proliferation. These processes take place in the nucleous where rRNA is assembled into ribosomal subunits. Increased rRNA synthesis and enlarged nucleous are among the most important molecular alterations in cancer cells. Pol I transcription machinery is efficiently regulated, responding to a variety of external signals such as the nutrient availability, growth factors, and cellular stress. Understanding this regulation is required for the development of new strategies for the molecular characterization and subsequent therapy of cancer.
In their review article by Harold and colleagues summarize the molecular mechanisms that control rRNA synthesis and discuss design and development of novel drugs to combat breast cancer through targeted downregulation of Pol I transcription. This is a nice and timely review that gives the perspective for the breast cancer treatment.
I have some only some minor comments concerning the text and figures
- Nucleous is membraneless organelle (as stated in the line 75). Why it is surrounded by a double line on Figure 2? This makes the impression that nucleous in fact has the membrane. Moreover, nucleolar import of the TTF-I termination factor is mentioned in the line 282. Please correct the figure and text to avoid misunderstanding.
- Figure 2 shows the complex regulation of Pol I transcription machinery, not dysregulation in breast cancer, as described in the legend.
- As stated in the legend to Figure 2, the 47S pre-rRNA transcript is subsequently processed and assembled with ribosomal proteins and the mature 5S rRNA to form the pre-40S and pre-60S subunits. Mature rRNA species were, however, not shown on the figure.
- Line 227: why PAF53 is called “important subunit of RNAPI”? Is it more important than the other subunits?
- Line 252: please explain what is the Bop1 protein
- Starting form unclear statement “the phosphate and tensin homolog” the whole “PTEN” paragraph is difficult to read and needs re- I assume, only partial deletion of PTEN is allowed. I have also trouble in understanding the meanings of “27 studies” , “triple-negative subtype” , “inhibition of dysregulated proliferation”.
- Description of p14Arf activity (lines 279-281). I guess, it reduces UBF1 activity and rDNA transcription by inhibition of UBF1 pghosphorylation at serines S484 and S388.
- The title of paragraph 5 should be “Regulation of rRNA synthesis by noncoding RNAs”
- What is Top1/2 on figure 3? It is not described in the legend and text.
- Lines 34 and 423: pre rRNA 47 S instead 47S
